# Percutaneous Chevron/Akin (PECA) versus open scarf/Akin (SA) osteotomy treatment for hallux valgus: A systematic review and meta-analysis

Gabriel Ferraz Ferreira[ORCID][1][°], Vinícius Quadros Borges[2][°], Leonardo Vinícius de Matos Moraes[3], Kelly Cristina Stéfani[4]*

1 Foot and Ankle Surgery Group, Orthopedics and Traumatology Department, Instituto Prevent Senior, São Paulo, São Paulo, Brazil, 2 Foot and Ankle Surgery Group, Orthopedics and Traumatology Department, Hospital do Campo Limpo, São Paulo, São Paulo, Brazil, 3 Foot and Ankle Surgery Group, Orthopedics and Traumatology Department, Hospital do Servidor Público Estadual, São Paulo, São Paulo, Brazil, 4 Centro de Inovação Tecnológica do Instituto Central, InovaHC, Hospital das Clínicas de São Paulo, São Paulo, São Paulo, Brazil

° These authors contributed equally to this work.
* kelly.stefani@hc.fm.usp.br

**Data Availability Statement:** All relevant data are within the manuscript and its Supporting information files.

## Abstract

### Purpose

The objective of the study is to compare the radiographic and clinical results of two techniques for the treatment of hallux valgus that have the same indication, the open scarf/Akin (SA) technique and the percutaneous Chevron/Akin (PECA).

### Methods

A meta-analysis was performed with the studies found during a systematic review of articles included in electronic databases until 30 May 2020. The pooled analysis was summarized according to clinical outcomes, such as visual analog pain scale (VAS) and American Orthopaedic Foot & Ankle Society (AOFAS) score, radiographic outcomes and complications, with a 95% confidence interval.

### Results

Three studies comparing the open scarf/Akin (SA) versus the PECA techniques were added to the analysis, corresponding to 235 feet, 102 in the PECA group and 133 in the SA. The final mean difference in the hallux valgus angle was 0.80 degrees and in the intermetatarsal angle 0.53, in the last radiographic evaluation. In the AOFAS score, the final mean difference was 4.97 points and in the VAS 0.14 in relation to the last clinical evaluation. Exposure to radiation during the surgical procedure was higher in the PECA group with a mean of 35.53 seconds.

**Funding:** The authors did not receive specific funding for this work.

**Competing interests:** The authors have declared that no competing interests exist.

## Conclusions

The PECA surgical technique for the treatment of hallux valgus when compared with SA demonstrated similar radiographic correction, pain and function after six months of follow-up but with a longer radiation exposure time.

## Register of systematic review (PROSPERO)

CRD42018096613.

## Introduction

Hallux valgus is a widely studied deformity that affects the feet, and even though more than 130 conventional techniques have been described for its correction, the best technique has not yet been determined [1].

One of the most commonly used techniques is open scarf/Akin (SA) osteotomy, which is performed through a medial approach in the forefoot and uses screw fixation to achieve absolute stability [2]. The results described are satisfactory; however, residual complains are reported in open surgery by approximately 15% of patients and include intense postoperative pain, slow recovery and stiffness [3–6].

This result led to a search for research on alternative methods for hallux valgus correction, and the concepts of minimally invasive surgery have been employed, increasing the number of publications on the subject in recent years [7,8].

Currently, the third generation of minimally invasive surgery for hallux valgus is the percutaneous Chevron/Akin (PECA) technique, described by Vernois and Redfern [9] as MICA (minimally invasive Chevron-Akin).

The reviews published so far suggest that there is a lack of randomized clinical trials comparing the PECA technique with conventional open techniques, as only one case series with low methodological quality was found in the literature [10,11].

The objective of this study was to carry out a systematic review with the following characteristics:

- Population: patients diagnosed with hallux valgus;

- Intervention: PECA;

- Control: SA;

- Outcome: pain, function and radiographic evaluation;

- Studies: randomized controlled trials or controlled retrospective studies;

## Methods

### Search strategy

A systematic review was performed by two reviewers according to the Preferred Reporting Items for Systematic Reviews and Meta-Analyses guidelines [12]. Studies were located by searching the Medline (PubMed), Cochrane Library, Scopus and Embase databases. The search was performed on 30 May 2020 with the keywords "hallux valgus AND (percutaneous

OR minimally invasive)", without any language restriction or filter. In addition, a manual search was performed of the references cited in studies, letters, reviews and foot and ankle reference textbooks. The present systematic review was registered in the International Prospective Register of Systematic Reviews (PROSPERO) [13]. Two reviewers retrieved the data and independently analyzed each selected study; instances of disagreement were resolved by the senior investigator.

## Inclusion and exclusion criteria

The inclusion criteria were the following: (1) hallux valgus diagnosis and (2) surgical treatment by the SA and PECA technique. The exclusion criteria were the following: (1) patients with previous treatments and (2) patients with associated neuromuscular diseases. Case reports, letters to the editor, systematic reviews and opinion pieces were removed.

## Data extraction

The article data were extracted by two independent researchers according to a previously established protocol. The data collected were tabulated according to the relevant data using a previously defined protocol.

## Quality assessment

We used two different methods for analyzing the quality of the studies (Table 1). For the study by Lai et al. [14] and Frigg et al. [15] we used ROBINS I [16] which classified the study as having a serious risk of bias and moderate risk of bias, respectively. In turn, the Revised Cochrane risk-of-bias tool for randomized trials (RoB 2.0) [17] was used to evaluate the study by Lee et al. [18] since it was a randomized controlled trial. The study was categorized as having a some concerns.

## Statistical analysis

The studies included were qualitatively analyzed, and the summary of their results is reported in tables. The meta-analysis was performed through the *Meta* package for R. The standardized mean difference and the mean difference were used for the continuous outcome variables. The dichotomous variables were compared using relative risk. The results were described with the corresponding 95% confidence interval (95% CI). A $p$ value $< 0.05$ was considered statistically significant. Heterogeneity among studies was calculated using the $I^2$ and $\tau^2$ statistics.

**Table 1.** A; Summary of risk of bias assessment for randomized controlled trials. B; Summary of risk of bias assessment for non-randomized controlled trials.

A

| Study | Randomization process | Deviations from intended | Missing outcome data | Measurement of the outcome | Selection of the reported result | Overall Bias | | |
|---|---|---|---|---|---|---|---|---|
| Lee et al | Low risk | Low risk | Some concerns | Some concerns | Some concerns | Some concerns | | |

B

| Study | Bias due to confounding | Bias in selection of participants into the study | Bias in classification of interventions | Bias due to deviations from intended interventions | Bias due to missing data | Bias in measurement of outcomes | Bias in selection of the reported result | Overall |
|---|---|---|---|---|---|---|---|---|
| Lai et al | Moderate risk of bias | Serious risk of bias | Low risk of bias | Serious risk of bias | No information | Serious risk of bias | Moderate risk of bias | Serious risk of bias |
| Frigg et al | Low risk of bias | Moderate risk of bias | Low risk of bias | Moderate risk of bias | Moderate risk of bias | Moderate risk of bias | Moderate risk of bias | Moderate risk of bias |

### Date items

The results gathered from the various databases were synthesized and categorized using End-Note X7.7.1 (Thomson Reuters, CA, USA). Duplicates were removed.

### Outcomes and prioritization

The selected outcomes included visual analog scale of pain, function according to the American Orthopaedic Foot & Ankle Society Hallux Metatarsophalangeal-Interphalangeal (AOFAS Hallux MTP-IP) [19] rating system, visual analog scale of pain, radiographic evaluation, quality of life assessment (Bonney and Macnab) [20] and satisfaction with the outcome. In addition, other outcomes, such as complications, radiation exposure time and surgery time, were evaluated.

## Results

### Study selection

The search retrieved 691 articles: 189 from Medline (PubMed), 267 from Embase, 22 from the Cochrane Library and 213 from Scopus. After excluding duplicates and articles not relevant to the topic, 27 articles were selected for full-text review. Finally, three studies met the inclusion criteria and were included in the qualitative analysis and meta-analysis. A flowchart representing study selection is shown in Fig 1.

### Study characteristics

The most recent study was published by Frigg et al. [15] in 2019 in which the authors compared the prospective results of the treatment of hallux valgus using the PECA versus open SA technique without randomization from January 2014 to December 2017. Fifty patients were included in the open group and 48 in the percutaneous group.

The article by Lee et al. [17] was published in 2017, in which the author conducted a randomized controlled trial between April 2012 and May 2015. Fifty patients were included, 25 in the PECA group and 25 in the SA group.

The study by Lai et al. [14] was a retrospective analysis of prospective data. The PECA and SA control groups included 29 and 58 feet, respectively, in a 2:1 ratio. The surgical procedures were performed between 2013 and 2014.

All studies identified in the systematic review reported that surgical hallux valgus correction was only indicated after failure of conservative treatment. In total, 235 feet were submitted to PECA (102 feet) or SA (133 feet) surgery. The main characteristics of the three studies are shown in Table 2.

Preoperative and postoperative clinical evaluations were performed using a visual analog pain scale, as shown in Table 3.

Another important clinical evaluation that was performed in all studies was the AOFAS score. This score takes into account the function, range of motion, pain and gait of the patient. The AOFAS score was applied both in the preoperative and in the immediate and late postoperative periods (Table 4).

Moreover, only the study by Lee et al. [17] used the Bonney and Macnab criteria to assess quality of life, while studies by Lai et al. [14] e de Frigg et al. [15] evaluated the degree of satisfaction with the procedure.

Radiographic evaluation was performed in all studies. The metatarsophalangeal angle, known as the hallux valgus angle (HVA), and the intermetatarsal angle (IMA) were measured

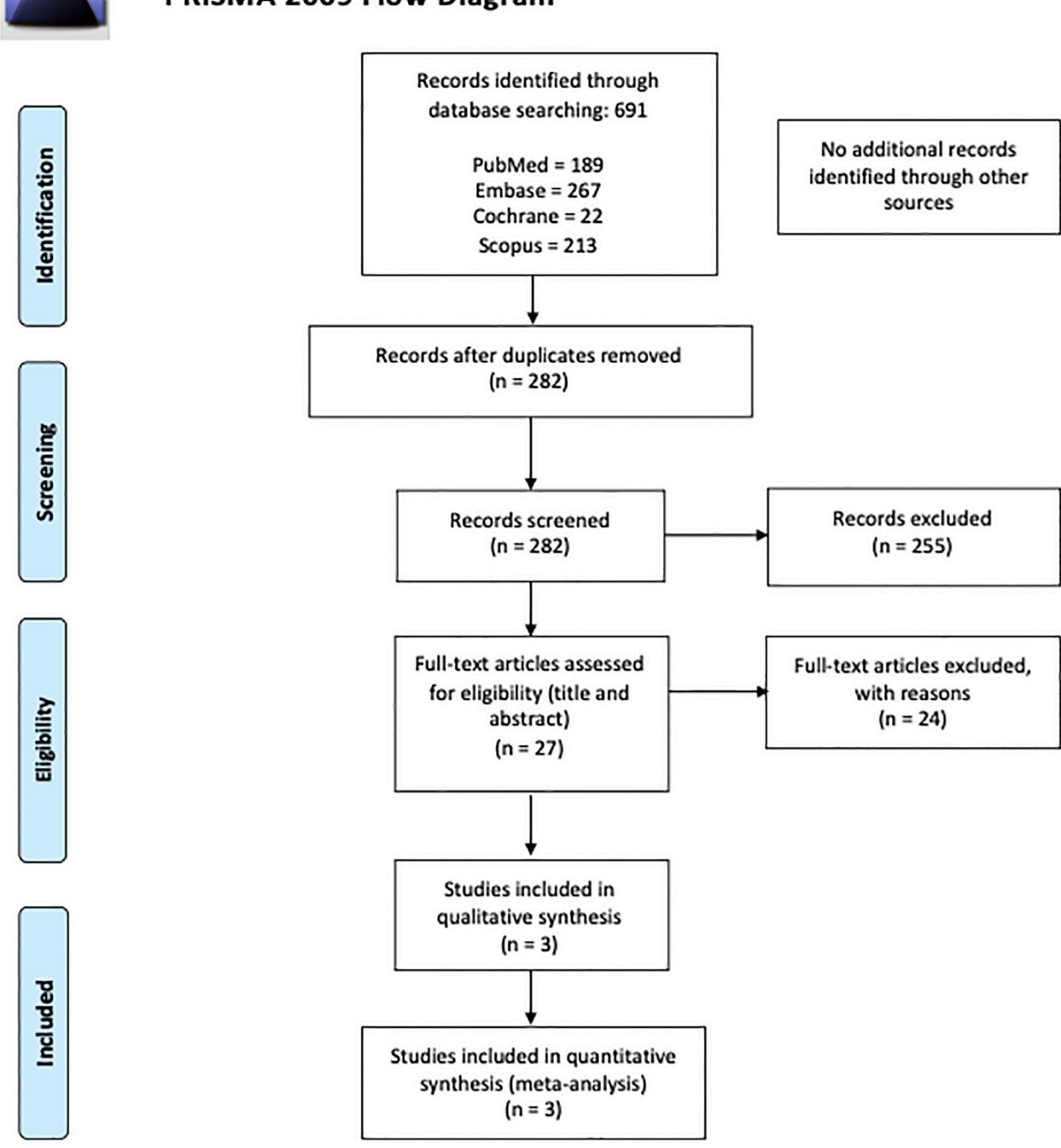

**Fig 1. PRISMA flowchart of the literature search and study selection.**

before the procedure. The last HVA and IMA measurements were considered the postoperative outcome. A summary of the radiographic evaluation results is shown in Table 5.

Other important characteristics were measured, such as radiation exposure time, surgical time, surgical scar size and complications. The PECA group presented complications only

**Table 2. Summary of studies included in the systematic review.**

| Study | Year of publication | Country | Study design | Age (mean) | | p | PECA | | | SA | | | p |
|---|---|---|---|---|---|---|---|---|---|---|---|---|---|
| | | | | PECA | SA | | Men | Woman | Total | Men | Woman | Total | |
| Lee et al | 2017 | Australia | Cohort study | 52.6 (20–76) | 53.4 (25–75) | 0.759 | 2 | 23 | 25 | 3 | 22 | 25 | 0.799 |
| Lai et al | 2017 | Singapore | Randomized controlled trial | 54.3 ± 12.8 | 54.3 ± 12.7 | 0.986 | 4 | 25 | 29 | 6 | 52 | 58 | 0.725 |
| Frigg et al | 2019 | Switzerland | Cohort study | 48.04* | 48.23* | 0.79 | 7 | 41 | 48 | 6 | 46 | 50 | 0.47 |

* Median.

**Table 3. Summary of evaluations by visual analog pain scale (VAS).**

| Study | VAS Pre | | △VAS Pre SA—PECA | p |
|---|---|---|---|---|
| | PECA | SA | | |
| Lee et al. | 7.1 ± 1.5 | 6.9 ± 1.7 | - 0.2 | 0.179 |
| Lai et al. | 4.0 ± 2.9 | 4.9 ± 2.6 | 0.9 | 0.124 |
| Frigg et al. | 3 (3, 4)* | 4 (3, 4)* | 1 | 0.35 |
| Study | VAS 1 day (Perioperative) | | △VAS 1 day (Perioperative) SA—PECA | p |
| | PECA | SA | | |
| Lee et al. | 2.2 ± 1.2 | 3.9 ± 1.9 | 1.7 | < 0.001 |
| Lai et al | 1.9 ± 0.6 | 3.9 ± 1.0 | 2 | < 0.001 |
| Frigg et al. | N/S | | | |
| Study | VAS 2 weeks | | △VAS 2 weeks SA—PECA | p |
| | PECA | SA | | |
| Lee et al. | 1.0 ± 1.4 | 2.4 ± 1.7 | 1.4 | < 0.001 |
| Lai et al. | N/S | | | |
| Frigg et al. | N/S | | | |
| Study | VAS 6 weeks | | △VAS 6 weeks SA—PECA | p |
| | PECA | SA | | |
| Lee et al. | 0.6 ± 1.8 | 2.1 ± 2.0 | 1.5 | 0.004 |
| Lai et al. | N/S | | | |
| Frigg et al. | N/S | | | |
| Study | VAS 6 months | | △VAS 6 months SA—PECA | p |
| | PECA | SA | | |
| Lee et al. | 0.3 ± 0.9 | 0.5 ± 1.1 | 0.2 | 0.160 |
| Lai et al. | 0.7 ± 1.8 | 0.9 ± 1.8 | 0.2 | 0.572 |
| Frigg et al. | N/S | | | |
| Study | VAS Last follow-up | | △VAS Last follow-up SA—PECA | p |
| | PECA | SA | | |
| Lee et al. | 0.3 ± 0.9 | 0.5 ± 1.1 | 0.2 | 0.160 |
| Lai et al. | 0.7 ± 1.9 | 0.4 ± 1.5 | -0.3 | 0.620 |
| Frigg et al. | 0 (0, 0)* | 0 (0, 0)* | 0 | 0.39 |

N / S: not specified;

* Used the median and interquartile range.

**Table 4. AOFAS score for pre- and postoperative functional evaluation.**

| Study | AOFAS Pre | | △AOFAS Pre (SA—PECA) | p |
|---|---|---|---|---|
| | PECA | SA | | |
| Lee et al. | 61.3 ± 3.2 | 58.5 ± 4.3 | -2.8 | 0.220 |
| Lai et al. | 58.6 ± 16.6 | 53.2 ± 14.6 | -5.4 | 0.127 |
| Frigg et al. | 52 (47, 60)* | 49 (44, 57)* | -3* | 0.06 |
| Study | AOFAS 6 months | | △AOFAS 6 months (SA—PECA) | p |
| | PECA | SA | | |
| Lee et al. | 88.7 ± 2.1 | 83.0 ± 3.5 | - 5,7 | 0.560 |
| Lai et al. | 85.6 ± 14.9 | 82.7 ± 14.5 | -2.9 | 0.183 |
| Study | AOFAS Last follow-up | | △AOFAS Last follow-up (SA—PECA) | p |
| | PECA | SA | | |
| Lee et al. | 88.7 ± 2.1 | 83.0 ± 3.5 | - 5,7 | 0.560 |
| Lai et al. | 87.4 ± 17.8 | 88.4 ± 13.8 | 1.0 | 0.547 |
| Frigg et al. | 95 (90, 100)* | 100 (86, 100)* | 5* | 0.60 |

* Used the median and interquartile range.

with the screws, in some cases being necessary to remove them. The SA group presented complications ranging from metatarsalgia to wounds complications.

Regarding the length of the surgical wound, it was measured by the study of Lee et al [18] with an average of 25.3mm in the PECA group and 107.1mm in the SA group, as well as in the study of Frigg study being the median of 5 cm for open surgeries and 1.2cm percutaneous (p < 0.001). In two studies there was the evaluation of radiation exposure that surgeons undergo during the procedure, being much higher in percutaneous surgeries than in open surgery. The operative time was measured only by the article by Lai et al [14], being greater in the procedures performed by the SA technique than by the PECA.

## Pooled analysis

The radiographic evaluation (IMA and HVA), clinical evaluation (AOFAS and VAS), complication and radiation exposure were analyzed together.

**Table 5. Summary of radiographic evaluations.**

| Study | HVA Pre | | p | HVA Last follow-up | | p |
|---|---|---|---|---|---|---|
| | PECA | SA | | PECA | SA | |
| Lee et al. | 31.4 ± 2.1 | 31.2 ± 4.1 | 0.890 | 7.6 ± 1.2 | 10.1 ± 1.9 | 0.520 |
| Lai et al. | 29.9 ± 8.5 | 30.6 ± 8.4 | 0.702 | 8.8 ± 5.9 | 13.8 ± 7.6 | 0.003 |
| Frigg et al. | 25 (21, 27)* | 25 (20, 30)* | 0.72 | 7 (5, 11)* | 10 (6, 14)* | 0.07 |
| Study | IMA Pre | | p | IMA Last follow-up | | p |
| | PECA | SA | | PECA | SA | |
| Lee et al. | 15.6 ± 1.0 | 15.7 ± 1.4 | 0.960 | 6.4 ± 0.8 | 7.6 ± 0.9 | 0.270 |
| Lai et al. | 14.6 ± 3.9 | 14.6 ± 3.3 | 0.954 | 10.3 ± 3.1 | 8.8 ± 3.4 | 0.055 |
| Frigg et al. | 13 (11, 14)* | 13 (11, 15)* | 0.93 | 6 (5, 8)* | 5 (4, 7)* | 0.008 |

* Used the median and interquartile range.

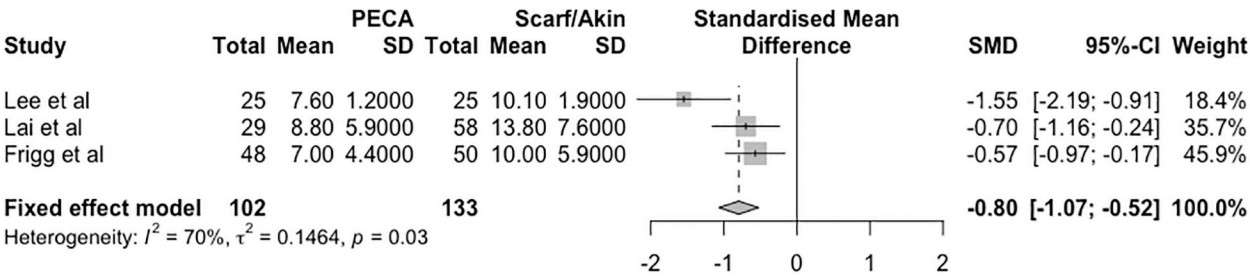

**Fig 2. Forest plot of the meta-analysis of the studies evaluating the final radiographic difference in the HVA.**

## Radiographic evaluation

The radiographic analysis of the studies was conducted based on two distinct measurements: the HVA and IMA at the last visit. Regarding the HVA found, there was a mean difference (HVA PECA—HVA SA) of -0.80 degrees in the fixed effects model (95% CI = -1.07 to -0.52, $p = 0.03$, $I^2 = 70\%$, $\tau^2 = 0.14$), as shown in Fig 2.

There was a difference between the mean IMA at the last radiographic evaluation (IMA PECA—IMA SA) of -0.53 degrees in the fixed effects model (95% CI = -0.93 to -0.13, $p < 0.01$, $I^2 = 93\%$, $\tau^2 = 2.37$), according to the forest plot in Fig 3.

## Clinical evaluation

Functional assessment was performed using the AOFAS score determined at the last clinical evaluation. The mean difference between the groups (PECA—SA) was 4.97 points in the fixed effects model (95% CI = 3.55 to 6.39, $p = 0.14$, $I^2 = 48\%$, $\tau^2 = 2.87$), as shown in the forest plot in Fig 4.

The visual analog pain scale was used postoperatively at two time points, the first of which was the first day after surgery, as shown in Fig 5, and the mean difference between the groups (PECA—SA) was -1.68 points in the fixed effects model (95% CI = -2.09 to -1.27, $p < 0.01$, $I^2 = 87\%$, $\tau^2 = 0.60$), performed only by two studies.

The analogue pain scale was also used at the last medical appointment, described in the three studies. The mean difference in the score between the groups (PECA—SA) was -0.14 points in the fixed effects model (95% CI = -0.49 to 0.20, $p = 0.81$, $I^2 = 0\%$, $\tau^2 = 0$), as shown in Fig 6.

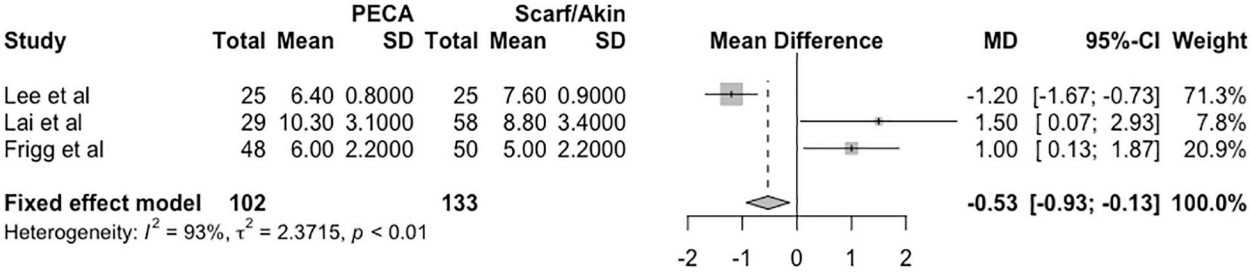

**Fig 3. Forest plot of the meta-analysis of the studies evaluating the final radiographic difference in the IMA.**

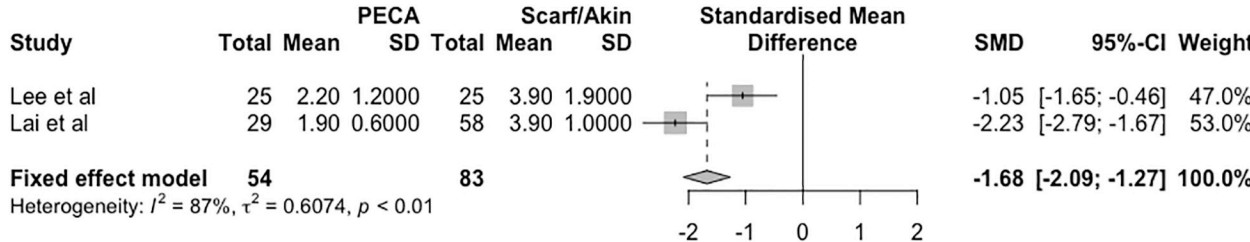

**Fig 4. Forest plot of the meta-analysis of the studies evaluating the difference in the AOFAS score at six months postoperatively.**

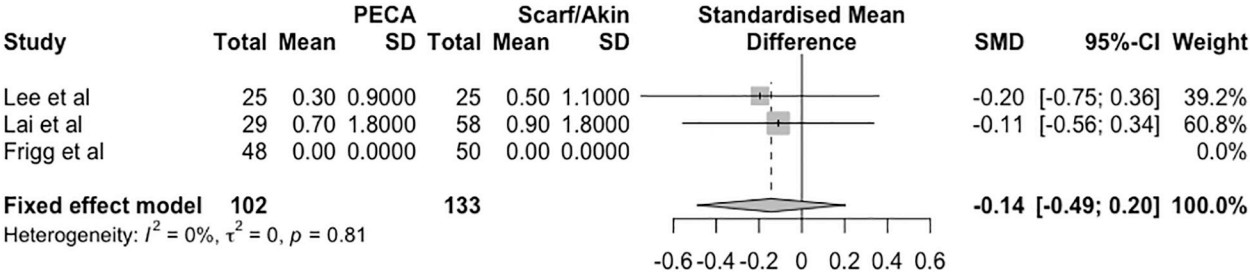

**Fig 5. Forest plot of the meta-analysis of the studies evaluating the difference in the visual analog pain scale score at one day postoperatively.**

## Complications

Complications in the groups are described in Fig 7 and were evaluated using the fixed effects model with a relative risk of 1.51 (95% CI = 0.80 to 2.86, $p = 0.36$, $I^2 = 3\%$, $\tau^2 = 0.01$). The complication was the removal of synthesis material, metatarsalgia and problems with wound healing.

## Radiation exposure

Exposure to radiation during the surgical procedure was assessed using the mean fluoroscopy duration in seconds. Fig 8 shows that the difference was 35.53 seconds between the PECA group and the SA group in the fixed effects model (95% CI = 31.75 to 35.31, $p < 0.01$, $I^2 = 87\%$, $\tau^2 = 11.31$).

| Study | Total | PECA Mean | SD | Total | Scarf/Akin Mean | SD | Standardised Mean Difference | SMD | 95%-CI | Weight |
|---|---|---|---|---|---|---|---|---|---|---|
| Lee et al | 25 | 0.30 | 0.9000 | 25 | 0.50 | 1.1000 | | -0.20 | [-0.75; 0.36] | 39.2% |
| Lai et al | 29 | 0.70 | 1.8000 | 58 | 0.90 | 1.8000 | | -0.11 | [-0.56; 0.34] | 60.8% |
| Frigg et al | 48 | 0.00 | 0.0000 | 50 | 0.00 | 0.0000 | | | | 0.0% |
| Fixed effect model | 102 | | | 133 | | | | -0.14 | [-0.49; 0.20] | 100.0% |

Heterogeneity: $I^2 = 0\%$, $\tau^2 = 0$, $p = 0.81$

**Fig 6. Forest plot of the meta-analysis of the studies evaluating the difference in the visual analog pain scale score at the last clinical evaluation.**

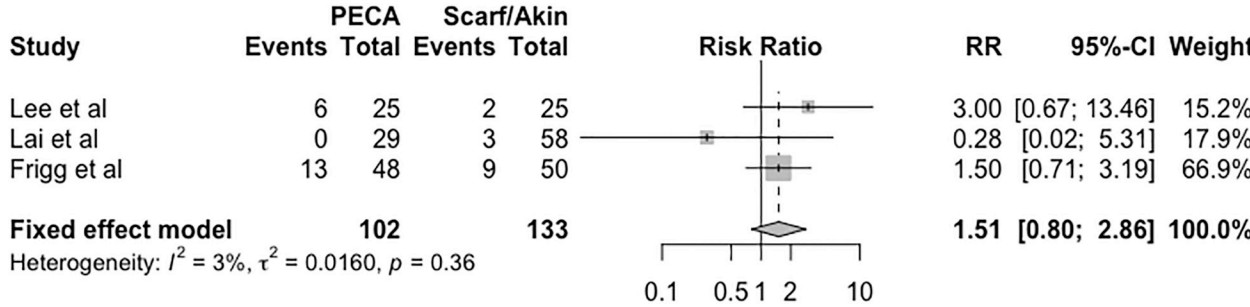

**Fig 7. Forest plot of the meta-analysis of the studies evaluating the relative risk of complications.**

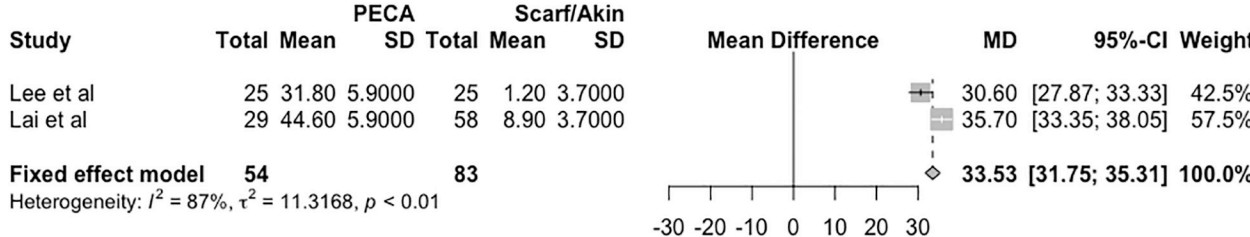

**Fig 8. Forest plot of the meta-analysis of the studies evaluating the difference in exposure to intraoperative radiation between the types of surgical procedures.**

## Publication bias

The funnel plot and Begg's test were not performed since the inclusion of only three studies in the meta-analysis does not allow this type of evaluation.

## Discussion

Although the techniques used for hallux valgus correction have been described and are familiar to foot and ankle surgeons, comparison between studies is not easy because of the wide variations in techniques and heterogeneous samples, as described in the systematic review by Bia et al. [10], in addition to the low methodological quality of the studies without control or case series, as found in other reviews [7,11].

Therefore, our study prioritized the search for articles on specific and widely used techniques such as SA in open surgery and PECA in percutaneous surgery, thus enabling comparison between the studies and a search for the best evidence regarding the outcomes.

In open surgeries for hallux valgus correction, it makes no sense to compare the Chevron/Akin technique with Scarf/Akin. However, the Percutaneous Chevron (PECA) technique is used for moderate or severe hallux valgus cases, with a similar indication for the open Scarf/Akin technique. In the radiographic evaluation, there was a difference between the groups in the final HVA and IMA, with means derived from the meta-analysis of 0.80 and 0.53 degrees, respectively. Radiographic correction was achieved in both groups, with a small difference between the groups. Although radiographic alignment did not necessarily guarantee clinical correction, both techniques demonstrated the potential for deformity correction.

In the studies included in the meta-analysis, the AOFAS score was used to measure the postoperative outcome, as was the visual analog pain scale score. The AOFAS score was higher

in the PECA group, with a small mean difference of 4.97 points. The visual analog pain scale was used at two time points: immediately postoperatively (only in two studies) and last clinical evaluation. At the first time point, there was a difference between the means of the groups, with the PECA group having a mean score that was 1.68 points lower. At last rating the difference between the groups decreased, with a mean difference of 0.14 points.

The theoretical advantages of percutaneous surgery include lower rates of morbidity and faster recovery with immediate full weight-bearing, which have led to an increased use of the technique [21]. Based on the results of the meta-analysis, the difference between the techniques favored PECA, which yielded a lower pain score in the visual analog scale. However, the result found was based on only two studies, and should be generalized with limitations. In addition, the scar size was only measured in the study of Lee et al. [17], and the mean for the PECA group (23.3 mm) was much lower than that for the SA group (107.1 mm).

Another important difference between the percutaneous surgical technique and the open technique is the radiation exposure time. Percutaneous surgery requires localization by fluoroscopy, while in the open technique, fluoroscopy is used only at the end of the procedure to check the final outcome and screw sizes. Therefore, both studies evaluated the radiation exposure time, and the mean for the PECA group was higher than that for the SA group, with a difference of 33.53 seconds. Thus, the greater use of fluoroscopy and its potential for radiation exposure can be considered negative features of the method.

The percutaneous techniques presented complications such as osteonecrosis, malunion, relapse and pseudoarthrosis [10,21]. Open surgeries also present complications similar to those of percutaneous surgeries, such as those previously described [22,23]. In our study, complications were assessed as outcomes through the relative risk, but without significance.

Although the PECA technique is minimally invasive, percutaneous surgeries present a long learning curve, with a higher risk of complications compared with that attained by more experienced surgeons, requiring training on cadavers and previous specific training [4]. Surgical time should also be taken into account. This parameter was only measured in the study by Lai et al. [14], and it was higher in the SA group (44.3 minutes) than in the PECA group (56.6 minutes).

This study has some limitations. Only three studies were included in the systematic review, two of which was retrospective. Another limitation was the short follow-up time in the studies, which could interfere with the outcome. These factors can influence the risk of bias and lead to erroneous conclusions.

## Conclusion

The PECA surgical technique for the treatment of hallux valgus when compared with SA demonstrated similar radiographic correction, pain and function after six months of follow-up but with a longer radiation exposure time. Only three studies were included in the meta-analysis. A multicenter study with the same design as that of the selected studies is necessary to confirm the results obtained.

## Supporting information

**S1 Checklist. PRISMA 2009 checklist.**
(DOC)

## Author Contributions

**Conceptualization:** Gabriel Ferraz Ferreira, Leonardo Vinícius de Matos Moraes, Kelly Cristina Stéfani.

**Data curation:** Gabriel Ferraz Ferreira, Leonardo Vinícius de Matos Moraes, Kelly Cristina Stéfani.

**Formal analysis:** Gabriel Ferraz Ferreira, Kelly Cristina Stéfani.

**Investigation:** Gabriel Ferraz Ferreira, Vinícius Quadros Borges.

**Methodology:** Gabriel Ferraz Ferreira, Vinícius Quadros Borges, Kelly Cristina Stéfani.

**Project administration:** Gabriel Ferraz Ferreira, Vinícius Quadros Borges, Kelly Cristina Stéfani.

**Resources:** Kelly Cristina Stéfani.

**Software:** Gabriel Ferraz Ferreira, Kelly Cristina Stéfani.

**Supervision:** Gabriel Ferraz Ferreira, Kelly Cristina Stéfani.

**Validation:** Gabriel Ferraz Ferreira, Leonardo Vinícius de Matos Moraes, Kelly Cristina Stéfani.

**Visualization:** Gabriel Ferraz Ferreira, Leonardo Vinícius de Matos Moraes, Kelly Cristina Stéfani.

**Writing – original draft:** Gabriel Ferraz Ferreira, Kelly Cristina Stéfani.

**Writing – review & editing:** Gabriel Ferraz Ferreira, Kelly Cristina Stéfani.

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
