## [Decision Letter · Decision Letter 0]

1 Sep 2020

PONE-D-20-20197

Percutaneous Chevron/Akin versus open scarf/Akin osteotomy treatment for hallux valgus: a systematic review and meta-analysis

PLOS ONE

Dear Dr. Stefani,

Thank you for submitting your manuscript to PLOS ONE. After careful consideration, we feel that it has merit but does not fully meet PLOS ONE’s publication criteria as it currently stands. Therefore, we invite you to submit a revised version of the manuscript that addresses the points raised during the review process.

We look forward to receiving your revised manuscript.

Kind regards,

Osama Farouk

Academic Editor

PLOS ONE

Journal Requirements:

2. Please ensure you have included the full electronic search strategy for at least one database and uploaded it as an additional file.

5. Please amend either the title on the online submission form (via Edit Submission) or the title in the manuscript so that they are identical.

Reviewers' comments:

Reviewer's Responses to Questions

**Comments to the Author**

1. Is the manuscript technically sound, and do the data support the conclusions?

Reviewer #1: Partly

Reviewer #2: Yes

2. Has the statistical analysis been performed appropriately and rigorously? 

Reviewer #1: Yes

Reviewer #2: Yes

3. Have the authors made all data underlying the findings in their manuscript fully available?

Reviewer #1: No

Reviewer #2: Yes

4. Is the manuscript presented in an intelligible fashion and written in standard English?

Reviewer #1: Yes

Reviewer #2: Yes

5. Review Comments to the Author

Reviewer #1: Thank you for asking me to review this paper dealing with a comparison between percutaneous Chevron/Akin and traditional Scarf/Akin. The topic is definitely interesting. I think the paper will be suitable for publication after a revision by authors. The most important point will be to remodulate the conclusion, since I don’t believe that data allow to say the early postoperative pain is better for PECA. It can be said that the outcomes are comparable after 6 months, but that no analysis is possible about the early recovery based on published data.

My comments in detail are:

ABSTRACT

I would recommend to include more data in the methods section. In results, how many patients were considered in the 3 studies in total?

INTRODUCTION

Line 59 Better to say The reviews published so far…

METHODS

Line 85 I think here you may want to say ‘…treatment by the SA and PECA technique »

RESULTS

VAS at 1 day is reported only by two studies and at 1 week and 2 weeks by one study

Line 170 Is this Portuguese?

Line 171 Typo

Line 179-180 This is not clear

Line 188 -0.53 what? and in favour of what technique?

Line 193 As above. Please, be clearer when reporting differences

Line 198 As above

Line 206 As above

Line 212 As above

DISCUSSION

Line 233 0.80 what? Please revise the whole paper indicating the Units for values

Line 237-243 I think this paragraph should be remodulated. Was the difference in AOFAS statistically significant? I think you cannot draw any conclusion on the immediate postoperative pain since only one or two studies assessed it.

Line 247 Pain score at VAS cannot be judged only on one or two studies

CONCLUSION

I think that, based on your numbers, you cannot infer that PECA led to less postoperative pain. You may say that the clinical outcome is comparable at 6 and 12 months, and that early recovery cannot be assessed based on current literature.

Reviewer #2: This study addresses a common problem in foot surgery. A well-tried technique is compared with a new percutaneous technique in the context of a meta-analysis. This analysis was carried out technically flawlessly. The only problem I see is that only 3 studies could be included. However, this fact was openly presented and discussed by the authors in the discussion.

Please delete „were excluded“ in line 87

6. PLOS authors have the option to publish the peer review history of their article (what does this mean?). If published, this will include your full peer review and any attached files.

Reviewer #1: No

Reviewer #2: **Yes: **PD Dr. med. habil. Kajetan Klos

---

## [Author Response · Author response to Decision Letter 0]

16 Oct 2020

Response to Reviewers

A detailed response to the recommendations of the reviewers is provided here, including the specific revisions made to the original manuscript. A table format has been used for clarity.

Reviewer 1

COMMENT RESPONSE TEXT CHANGES

General

Thank you for asking me to review this paper dealing with a comparison between percutaneous Chevron/Akin and traditional Scarf/Akin. The topic is definitely interesting. I think the paper will be suitable for publication after a revision by authors. The most important point will be to remodulate the conclusion, since I don’t believe that data allow to say the early postoperative pain is better for PECA. It can be said that the outcomes are comparable after 6 months, but that no analysis is possible about the early recovery based on published data. Thank you for this comment. We agree that the data is not robust enough for this conclusion. We changed the conclusion of the study as suggested by the reviewer. Thank you very much. 

Abstract

I would recommend to include more data in the methods section. In results, how many patients were considered in the 3 studies in total? Perfect. We added in abstract and results the total number of feet that underwent surgical treatment and the results found in the meta-analysis, valuing the data included in the three studies. The conclusion was modified according to the reviewer's suggestion. Line 14 and Line 122-123

Introduction

Line 59 Better to say The reviews published so far… The sentence really got confused. We changed it as suggested. Thank you. Line 37

Methods

Line 85 I think here you may want to say ‘…treatment by the SA and PECA technique » We agree with the comment. We changed the "or" to "and". It was much better that way. Thank you. Line 63

Results

Line 170 Is this Portuguese? Translated. This was a mistake by the translator. We apologize and appreciate the comment. Line 146

Line 171 Typo Adjusted. Line 147

Line 179-180 This is not clear The item was to be clear about subsequent analyzes. But it was loose in the text and confused. Thanks for the comment. We decided to remove. 

Line 188 -0.53 what? and in favour of what technique? The grouped analysis corresponds to the difference in the averages. We standardize the calculations to be PECA minus SA, in the text is for example "(HVA PECA - HVA SA)". If the difference in the average is negative as in the case -0.53, it means that the mean of SA is greater than PECA. This comparison in means is standardized for all forest plots in the present study, as in the other values found. 

Line 193 As above. Please, be clearer when reporting differences 

Line 198 As above 

Line 206 As above 

Line 212 As above 

Discussion

Line 233 0.80 what? Please revise the whole paper indicating the Units for values Thank you for the question. In this case 0.80 and 0.53 are the difference in the means of the HVA and IMA angles. “...final HVA and IMA, with means derived from ...” We added the word "degrees" in the sequence of values for easy reading. Lines: 161, 164, 169, 174, 178

Line 237-243 I think this paragraph should be remodulated. Was the difference in AOFAS statistically significant? I think you cannot draw any conclusion on the immediate postoperative pain since only one or two studies assessed it. Very good comment. We cannot affirm and conclude with absolute certainty the results, as we have points of possible bias in the study. We included in the last paragraph of the discussion: "These factors can influence the risk of bias and lead to erroneous conclusions." The result of the meta-analysis we interpret as a pooled analysis and heterogeneity (i2), and not on statistical significance. The difference in the means is a statistical synthesis that can present distortions that are demonstrated in the weak points of the study. 

Line 247 Pain score at VAS cannot be judged only on one or two studies In fact, the low number of articles included in the meta-analysis can increase the risk of bias and restrict its generalization. We add in this paragraph that the result was obtained from only two studies, with this evident limitation. We appreciate the suggestion. Line 223 - 224

Conclusion

I think that, based on your numbers, you cannot infer that PECA led to less postoperative pain. You may say that the clinical outcome is comparable at 6 and 12 months, and that early recovery cannot be assessed based on current literature. The conclusion was modified according to the reviewer's suggestion, both in the abstract and in the conclusion. Line 253 – 255 and Line 19 - 21

Reviewer 2

COMMENT RESPONSE TEXT CHANGES

This study addresses a common problem in foot surgery. A well-tried technique is compared with a new percutaneous technique in the context of a meta-analysis. This analysis was carried out technically flawlessly. The only problem I see is that only 3 studies could be included. However, this fact was openly presented and discussed by the authors in the discussion. We really appreciate your time for this review. We hope that after the modifications suggested by the reviewers, the article will be suitable for publication. Thank you very much. 

Please delete „were excluded“ in line 87 Thank you for the comment. The word "excluded" was really inappropriate. We changed it to "removed". Thank you. Line 65

---

## [Decision Letter · Decision Letter 1]

4 Nov 2020

Percutaneous Chevron/Akin (PECA) versus open scarf/Akin (SA) osteotomy treatment for hallux valgus: a systematic review and meta-analysis

PONE-D-20-20197R1

Dear Dr. Stefani,

We’re pleased to inform you that your manuscript has been judged scientifically suitable for publication and will be formally accepted for publication once it meets all outstanding technical requirements.

Kind regards,

Osama Farouk

Academic Editor

PLOS ONE

Additional Editor Comments (optional):

Reviewers' comments:

Reviewer's Responses to Questions

**Comments to the Author**

1. If the authors have adequately addressed your comments raised in a previous round of review and you feel that this manuscript is now acceptable for publication, you may indicate that here to bypass the “Comments to the Author” section, enter your conflict of interest statement in the “Confidential to Editor” section, and submit your "Accept" recommendation.

Reviewer #1: All comments have been addressed

Reviewer #2: All comments have been addressed

2. Is the manuscript technically sound, and do the data support the conclusions?

Reviewer #1: Yes

Reviewer #2: Yes

3. Has the statistical analysis been performed appropriately and rigorously? 

Reviewer #1: Yes

Reviewer #2: I Don't Know

4. Have the authors made all data underlying the findings in their manuscript fully available?

Reviewer #1: Yes

Reviewer #2: Yes

5. Is the manuscript presented in an intelligible fashion and written in standard English?

Reviewer #1: Yes

Reviewer #2: Yes

6. Review Comments to the Author

Reviewer #1: The authors have addresses all the concerns. I believe that the paper can be published in its current form.

Reviewer #2: (No Response)

7. PLOS authors have the option to publish the peer review history of their article (what does this mean?). If published, this will include your full peer review and any attached files.

Reviewer #1: No

Reviewer #2: No

---

## [Editor Report · Acceptance letter]

16 Dec 2020

PONE-D-20-20197R1 

Percutaneous Chevron/Akin (PECA) versus open scarf/Akin (SA) osteotomy treatment for hallux valgus: a systematic review and meta-analysis 

Dear Dr. Stefani:

I'm pleased to inform you that your manuscript has been deemed suitable for publication in PLOS ONE. Congratulations! Your manuscript is now with our production department. 

Kind regards, 

on behalf of

Dr. Osama Farouk 

Academic Editor

PLOS ONE